# Land Tenure, Loans, and Farmers' Cropland Conservation Behavior: Evidence from Rural Northwest China

**Guoren Long** [1], **Xiaoyan Zhou** [2] and **Jun Li** [2,*]

[1]  Institute of Central Asia, Shaanxi Normal University, Xi'an 710119, China; longguoren@snnu.edu.cn
[2]  Northwest Institute of Historical Environment and Socio-Economic Development, Shaanxi Normal University, Xi'an 710119, China; zhouxiaoyan@snnu.edu.cn
*  Correspondence: lijun@snnu.edu.cn

**Abstract:** The pivotal role of farmers' cropland conservation behavior (CCB) in advancing green agricultural practices is well-recognized. This paper underscores the critical role of stable land tenure in enhancing farmers' CCB, exemplified by the practice of mulch recycling. Drawing on a survey of 349 cotton farmers in Xinjiang, Northwest China, it offers a systematic examination of how land tenure stability influences CCB and its underlying mechanisms. The findings reveal a significant positive correlation between land tenure stability and CCB. Notably, this relationship is mediated by the facilitation of land mortgages, wherein written contracts and extended land tenure durations enhance farmers' participation in land mortgages, thereby bolstering CCB. Furthermore, the stabilizing effect of land tenure on CCB also mitigates the negative impacts of risk aversion and time preference. The study additionally highlights the differential effects of land tenure stability based on farm size and technical training; its facilitative role in CCB is more pronounced among larger-scale farmers and those engaged in technical training.

**Keywords:** land tenure; loans; cropland conservation; farm scale; technical training

## 1. Introduction

Cropland conservation behavior (CCB) is essential for sustainable agricultural development, yielding significant economic and environmental benefits. Economically, it bolsters crop yields and ensures long-term income stability for farmers. Environmentally, it is crucial for the protection of farmland [1–3]. A prime example of CCB is mulch recycling, which notably reduces soil and ecological pollution from mulch residue. In China, the leading consumer of plastic mulch film globally, responsible for approximately 60% of its use, such film fulfills various agricultural functions. These include enhancing ground temperature, water, soil, and fertilizer conservation, aiding in weed and pest control, preventing drought and waterlogging, suppressing salt, protecting seedlings, and improving near-ground light and heat conditions, all contributing to better product sanitation and cleanliness. However, despite its myriad advantages, the extensive use of plastic mulch film has led to considerable agricultural non-point-source pollution [4], posing a significant challenge to CCB.

The recycling of plastic film is an important CCB practiced by farmers. Plastic mulch film significantly contributes to agricultural productivity by enhancing soil warming, water retention, soil fertility, and reducing soil erosion, thus aiding in the prevention of agricultural disasters [5,6]. Its use is particularly prevalent in the arid regions of northwest China, notably in Xinjiang and Gansu [7]. As depicted in Figure 1, from 2000 to 2016, the application of plastic mulch film in China has shown a consistent annual increase, averaging a growth rate of 3.6%. However, between 2017 and 2022, the growth rate decelerated due to the implementation of policies aimed at reducing its use. The 2023 China Rural Statistical Yearbook reveals regional variations in plastic mulch film coverage across China, with the highest usage in the western region, followed by the central region. In 2023, Xinjiang

recorded the highest usage levels among all Chinese regions. However, the shift in the use of plastic mulch film from being hailed as a "white revolution" to being criticized as "white pollution"[1] has resulted in substantial environmental costs, particularly for developing countries. While the benefits of plastic mulch recycling include enhancing crop yield, conserving water resources, and boosting economic advantages [5,8,9], the implementation of plastic mulch recycling in China has progressed slowly. Data from the Ministry of Agriculture and Rural Affairs (2017) indicate that less than 70% of Chinese farmers have adopted plastic mulch recycling. The reasons for this are multifaceted. Firstly, plastic mulch film, primarily composed of polyethylene, is resistant to degradation. Its long-term usage leads to the accumulation of fragments in the soil, posing challenges for recycling [1]. However, the practices of burning or leaving plastic mulch film in the soil detrimentally impact plant growth and soil fertility [4,10]. Hu et al. [6] found that residues from plastic mulch film hinder crop root development, a negative effect that becomes more pronounced when residues surpass 300 kg/ha. It is evident that plastic film recycling is a vital technology for arable land protection, effectively managing residual pollution. Furthermore, farmers, as fundamental participants and stakeholders in CCB, significantly influence the success of sustainable agricultural practices.

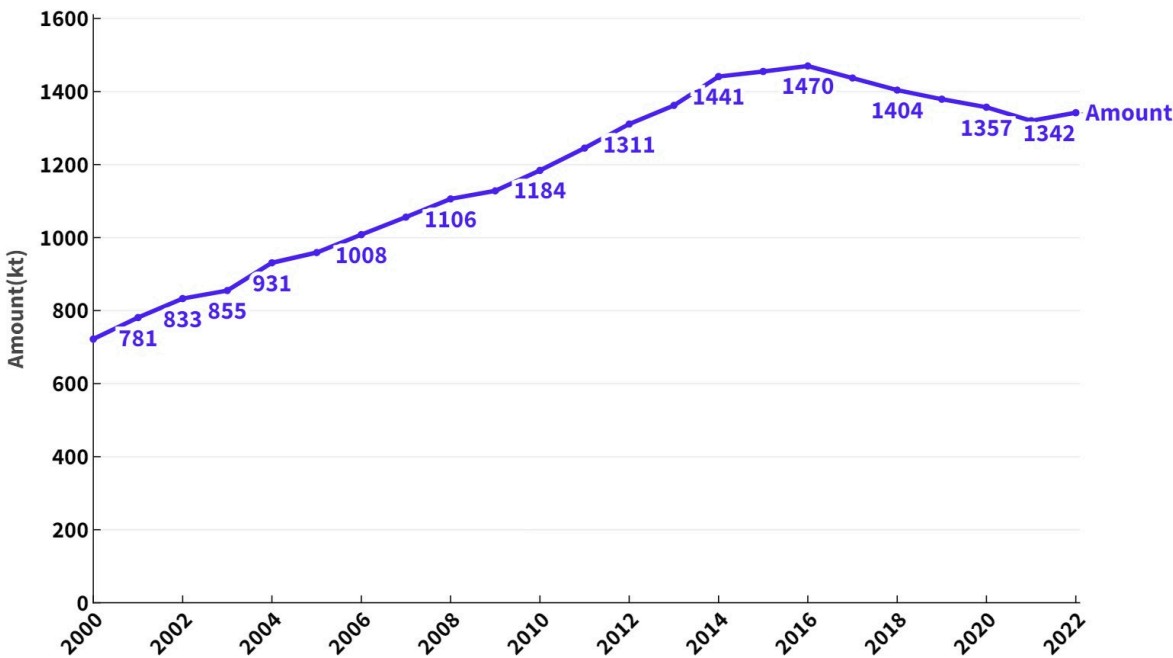

**Figure 1.** Utilization of plastic mulch film in China from 2000 to 2022[2].

To enhance the effectiveness of CCB, the Chinese government has enacted various policies, including the Central Committee's No.1 Document (2015–2021), which aims to establish a comprehensive system for collecting, utilizing, and treating agricultural waste, thereby promoting the sustainable use of agricultural resources. The "Technical Guidelines for Agricultural Green Development (2018–2030)", issued by the Ministry of Agriculture and Rural Affairs, explicitly emphasize enhancing the recycling and utilization of plastic mulch film and developing a technical framework for sustainable agriculture. However, a historical focus on usage over recycling has exacerbated plastic mulch film pollution in China. Despite recycling being crucial for preserving arable land quality, the national rate remains below 70% (Ministry of Agriculture and Rural Affairs, 2017). Consequently, exploring a long-term mechanism for farmers' CCB in China holds significant practical importance.

Research on CCB predominantly encompasses the following domains: Firstly, investigations have focused on the relationship between land tenure and CCB, where empirical evidence suggests that land tenure stability significantly enhances sustainable agricul-

ture [11] and encourages the adoption of eco-friendly agricultural technologies [3]. This stability also facilitates the widespread implementation of sustainable practices among farmers [12]. Secondly, the correlation between farm size and CCB has been explored. Larger farm sizes are associated with a greater inclination towards sustainable agricultural practices [13,14] and a higher likelihood of receiving credit from rural financial institutions, unlike their smaller counterparts, who face difficulties in securing loans [15,16]. Thirdly, studies have examined the role of time preference in CCB adoption, highlighting how the anticipation of future benefits and the uncertainty of outcomes influence farmers' decisions [17]. Lastly, research has addressed the impact of risk aversion on CCB, where a tendency towards uncertainty avoidance reduces the adoption rates of innovative agricultural techniques [18]. Additional factors identified include family income [19], age of the household head [20], and external elements such as economic policies [21], technical training [22], and information technology [23]. However, there is a notable lack of focus on how land tenure stability, particularly through land mortgage loans, influences CCB, considering variations in farm scale and technical training. Furthermore, the literature does not adequately explore the moderating effect of land tenure stability on farmers' risk aversion and time preferences in the context of CCB, thereby identifying the deficiency for this paper to make up.

Property rights theory[3] posits that well-defined and stable land property rights can mitigate investment risks for farmers, ease funding constraints, and lower transaction costs, thereby increasing the likelihood of CCB adoption [24,25]. The impact of land tenure stability on CCB adoption manifests in several ways. Firstly, it provides direct incentive effects by fostering long-term production planning [26], enhancing access to future profits, and motivating CCB application in agriculture [27]. Secondly, land tenure stability facilitates credit access, as it aids farmers in securing land mortgage loans [28], reduces loan acquisition costs, and eases funding challenges, promoting CCB adoption. Lastly, it influences transaction income effects: stable tenure simplifies land transfers, leading to land concentration among more productive farmers [29], thereby creating economies of scale. Households with larger-scale operations and more stable land rights face lower technological costs and focus more on long-term benefits, making them likelier to adopt CCB [13,30].

China offers a distinctive context for examining this issue. Firstly, in contrast to developed countries, Chinese farms are generally smaller in scale, and land tenure security is less assured [31], making the study of land tenure stability's impact on CCB adoption in China a valuable reference for other emerging economies. Secondly, as per "The 2019 National Bulletin on Cultivated Land Quality Grades", only 31.24% of China's total cultivated land is of excellent quality[4], with the majority facing challenges of suboptimal basic fertility. Thirdly, data from the 'China Rural Statistical Yearbook' (2020) reveal that China's use of plastic mulch film amounts to 1.379 million tons, covering an area of 176.281 million hectares, thereby making it the world's largest user by the end of 2019. Residues from plastic mulch film contribute to soil erosion, farmland soil degradation, and a decline in farmland productivity [32]. Consequently, CCB not only offers long-term profit potential for farmers but also facilitates green agricultural development by enhancing cultivated land quality [1].

This paper aims to investigate the potential positive impact of land tenure stability on farmers' CCB adoption. It further explores whether this effect varies with differences in farm size and farmers' involvement in technical training. Additionally, the study examines if land tenure stability moderates the impact of risk aversion and time preferences on CCB adoption.

Utilizing survey data from 349 cotton farmers in Xinjiang Uygur Autonomous Region (the following abbreviation is Xinjiang), China, in 2019, this study conducts a comprehensive analysis of how land tenure stability affects farmers' CCB adoption, particularly through the lens of land mortgage loans. Moreover, it examines the role of land tenure stability in moderating the effects of farmers' risk aversion and time preferences on their

CCB adoption. The study posits that the impact of land tenure stability on CCB adoption varies according to farm size and the extent of technical training.

The findings indicate that land tenure stability substantially enhances the probability of farmers adopting CCB. Essentially, more stable land tenure correlates with a higher likelihood of CCB adoption. This effect is primarily facilitated through land mortgage loans; specifically, written contracts and extended land tenure periods increase farmers' participation in these loans, thereby boosting their adoption of CCB. Furthermore, land tenure stability mitigates the negative impact of risk aversion and time preferences on CCB adoption. The influence of land tenure stability on CCB adoption varies with farm size and technical training, being more pronounced in large-scale farming operations and among farmers engaged in technical training.

This paper makes several key contributions to the existing literature. Firstly, it establishes that land tenure stability enhances farmers' CCB adoption, examining this through the lens of land mortgage loans, thereby enriching the understanding of critical factors influencing CCB adoption. Secondly, it investigates how land tenure stability moderates the impact of risk aversion and time preferences on CCB adoption. It posits that the influence of land tenure stability varies depending on farm size and technical training, offering a novel approach to increasing the low rate of CCB adoption among farmers. Finally, given Xinjiang's unique regional attributes, such as extensive land, sparse population, prevalent large-scale farming, and widespread land transfers, the study provides a distinctive sample. The paper takes into account Xinjiang's specific characteristics by focusing on cotton farmers, including ethnic minorities and corps cotton farmers, thereby presenting new and valuable data for research.

## 2. Theoretical Analysis and Research Hypotheses

Modern property rights theory posits that a well-defined and secure land property rights system is crucial for the effective allocation of agricultural production factors and the promotion of sustainable agricultural development [33,34]. An advanced and efficient land property rights system not only safeguards farmers' land rights and interests but also facilitates the optimal utilization of rural land resources and the achievement of agricultural modernization [35]. Research indicates that land tenure stability enhances farmers' expected long-term income, alleviates financial constraints, and lowers the costs associated with adopting green agricultural technology (GAT) [25]. These factors collectively motivate farmers to adopt CCB. Consequently, land tenure stability influences farmers' CCB adoption through direct incentives, credit availability, and transaction income improvements.

The primary mechanism involves direct incentive effects. Land tenure stability significantly motivates farmers to invest long-term in their land, thereby positively influencing CCB adoption. It fosters stable property rights expectations among farmers, ensuring that they can recover investment costs and potentially achieve higher returns [36], thus spurring the adoption of CCB. Research conducted across various contexts supports this relationship. Abdulai et al. in Ghana [37], Jacoby in China [38], Domeher in Africa [36], and Deininger et al. [39] in Ethiopia have all demonstrated that land tenure stability positively impacts farmers' GAT adoption behavior.

The second mechanism involves credit effects. Stable land tenure substantially lowers loan interest rates for farmers, enhances their loan acquisition capabilities, and increases their chances of securing funds via land mortgage loans [40]. While financial support is crucial for adopting CCB, farmers often encounter significant credit constraints in rural credit markets due to limited repayment capacity and lack of collateral [41]. Nevertheless, stable land tenure enhances farmers' credit accessibility [42]. It also aids in reducing credit evaluation and transaction costs, enabling farmers to leverage land assets as collateral for credit, thereby alleviating financial limitations and fostering CCB adoption.

The third mechanism involves transaction income effects. Stable land tenure enhances the mobility of production factors and resource endowment capabilities, consequently increasing farmers' income through facilitating land transfers [37,43]. This, in turn, encour-

ages the adoption of CCB. On one hand, land tenure stability diminishes uncertainty in land transactions, boosts farmers' engagement in land transfer markets, and simplifies land leasing or selling. This ability to secure investments through land transactions reduces long-term investment risks [44], further promoting CCB adoption. On the other hand, stable tenure leads to land concentration among more capable farmers, creating economies of scale. Large-scale farmers, typically realizing higher profits, are more inclined towards GAT [29,45], thus increasing the likelihood of adopting CCB.

Land tenure stability primarily facilitates farmers' adoption of CCB through three mechanisms: direct incentive effects, credit effects, and transaction income effects. As such, greater land tenure stability correlates with a higher likelihood of CCB adoption. Consequently, this leads to the following hypothesis:

**H1:** *Land tenure stability can promote farmers' CCB adoption.*

Land tenure stability reinforces farmers' land rights, enhances their security awareness, and thereby encourages the adoption of GAT [46]. However, its impact on CCB adoption is significantly modulated by farmers' risk aversion. CCB introduces varying levels of risk and elevates technology adoption costs, creating uncertainty around technical benefits and decelerating technology diffusion [47]. Consequently, more risk-averse farmers are less inclined to adopt CCB [48,49]. Nonetheless, stable land tenure allows farmers to access government agricultural subsidies and credit insurance, mitigating financial constraints [50] and enhancing risk awareness, thereby diminishing the hesitance of risk-averse farmers towards CCB adoption. Thus, enhanced land tenure stability increases the likelihood of farmers' land investment, subsequently reducing the impact of risk aversion on CCB adoption. Consequently, this paper proposes the following hypothesis:

**H2:** *Land tenure stability positively regulates the impact of risk aversion on farmers' adoption of CCB.*

Time preferences significantly impact farmers' decisions regarding the adoption of intertemporal technologies due to their inconsistent nature [51–53]. For example, farmers with a "short-sighted" approach prioritize immediate income, showing reluctance towards technologies like CCB, which offer long-term advantages but lack immediate benefits [45,54,55]. However, land tenure stability, by ensuring a stable land use period and decreasing the uncertainty of future investment returns, incentivizes farmers to invest in the long term, thereby encouraging CCB adoption [56]. Stronger land tenure stability leads to improved operational income guarantees for farmers, consequently mitigating the negative impact of time preferences on CCB adoption. This observation leads to the proposal of the following hypothesis:

**H3:** *Land tenure stability exerts a positive moderating effect on the impact of time preferences on farmers' adoption of CCB.*

Research indicates that large-scale farmers exhibit a significantly higher rate of technology adoption than their small-scale counterparts, a disparity largely due to differences in resource endowments [57]. Firstly, by expanding land holdings, large-scale farmers generate economies of scale, effectively reducing the unit cost of adopting green agricultural technology (GAT) and enhancing the likelihood of adopting CCB [13]. Secondly, with more stable land tenure, large-scale farmers are better positioned to secure the necessary funds for CCB through credit channels [31,38]. Thirdly, these farmers typically have higher agricultural incomes and lower discount rates, focusing more on long-term returns [20]. They are also more inclined to expand land scale via land consolidation, thereby addressing the issue of land fragmentation and reducing labor costs. This strategic approach allows them to spread risks and seize more opportunities to implement GAT [14,58]. Consequently, larger-scale farmers with stronger land tenure stability are more likely to enhance agricul-

tural output and increase income through CCB compared to small-scale farmers. This leads to the following hypothesis:

**H4:** *The influence of land tenure stability on CCB adoption is more pronounced among large-scale farmers compared to their small-scale counterparts.*

Technical training plays a crucial role in disseminating GAT knowledge, thereby enhancing farmers' awareness and comprehension of GAT, which in turn increases the likelihood of its adoption [22]. Specifically, it serves dual purposes: firstly, it offers guidance on relevant technical knowledge, assists farmers in assessing the long-term economic benefits of CCB, and enhances their understanding of CCB [59], thereby facilitating its adoption. Secondly, as a platform for information exchange, technical training fosters the sharing of CCB insights and bolsters farmers' expectations of technical benefits through successful demonstrations in specific communities [60]. Moreover, technical training exhibits a spillover effect, where information about CCB provided during training permeates from trained to untrained farmers, reducing overall technical learning costs [61] and thus increasing the likelihood of CCB adoption. Based on these considerations, the following hypothesis is formulated:

**H5:** *Land tenure stability plays a more significant role in enhancing the adoption of CCB among trained farmers compared to those who are untrained.*

In summary, the theoretical framework and research hypothesis are as shown in Figure 2.

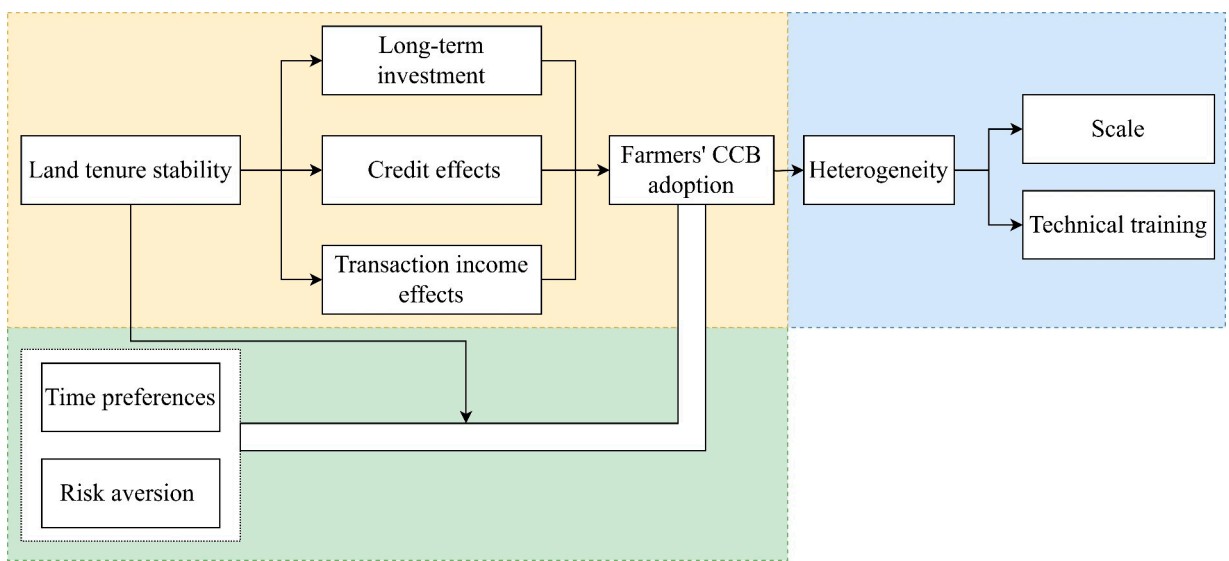

**Figure 2.** Theoretical framework and research hypothesis.

## 3. Material and Methods

### 3.1. Econometric Model

This study aims to investigate the influence of land tenure stability on farmers' adoption of CCB. Drawing on the frameworks of Ren et al. [62] and Li et al. [32], this paper identifies the adoption behavior of farmers' CCB as the dependent variable, with land tenure stability serving as the primary independent variable. The regression model employed is outlined below:

$$Y_1 = \alpha + \beta Tenure + \gamma X + \mu_1 \tag{1}$$

In the model (Formula (1)), $Y_1$ represents the dependent variable, denoting the adoption behavior of farmers' CCB. This is quantified by the extent of plastic mulch film recycling and the proportion of farmers engaging in CCB. *Tenure*, the primary independent variable, is assessed based on land tenure and lease type. *X* encompasses the control variables, which, following prior research, include household demographics, management characteristics, and soil quality indicators. These consist of the household head's age, gender, education level, farming experience, the number of household farm laborers, technical subsidies, corps farmer status, and historical cotton fertility. $\mu_1$ denotes a random error term, $\alpha$ represents a constant term, and $\beta$ and $\gamma$ are the coefficients to be estimated.

### 3.2. Data Collection

This study focuses on cotton farmers in Xinjiang as the subject of research for several key reasons. Firstly, within the framework of China's household contract responsibility system, the prevalent agricultural model involves high inputs, high outputs, and significant resource and environmental costs. This approach has led to an increasingly acute issue of cultivated land quality degradation [63]. Data from the Ministry of Agriculture and Rural Affairs of China reveals that in 2019, out of the 2.023 billion mu of surveyed and evaluated cultivated land, the average land quality was rated at 4.76. Notably, the cumulative area of medium-quality land reached 46.81%, with the land quality in the western region falling below the national average[5].

Secondly, cotton farmers in Xinjiang typically operate on a larger scale, with land transfer being a common phenomenon [64]. As of September 2020, the total area of land transferred in Xinjiang reached approximately 10.6577 million mu[6]. Thirdly, Xinjiang stands out as China's premier region for high-quality and high-yield cotton production, boasting the highest cotton-planting area in the country (refer to Figure 3). Data from the National Bureau of Statistics (2019) indicate that Xinjiang's cotton-planting area encompasses 38.1075 million mu, representing 76% of China's total cotton-planting area, with notable concentrations in the Kashi and Aksu regions (refer to Figure 4). In Xinjiang, cotton planting is a critical component of agricultural management[7].

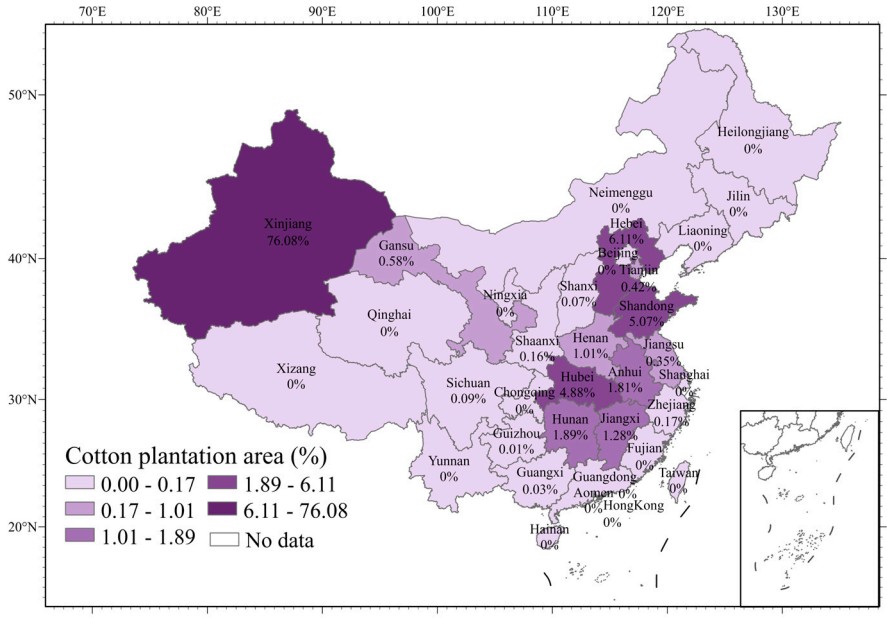

**Figure 3.** The proportion of cotton area planted in each city of China[8].

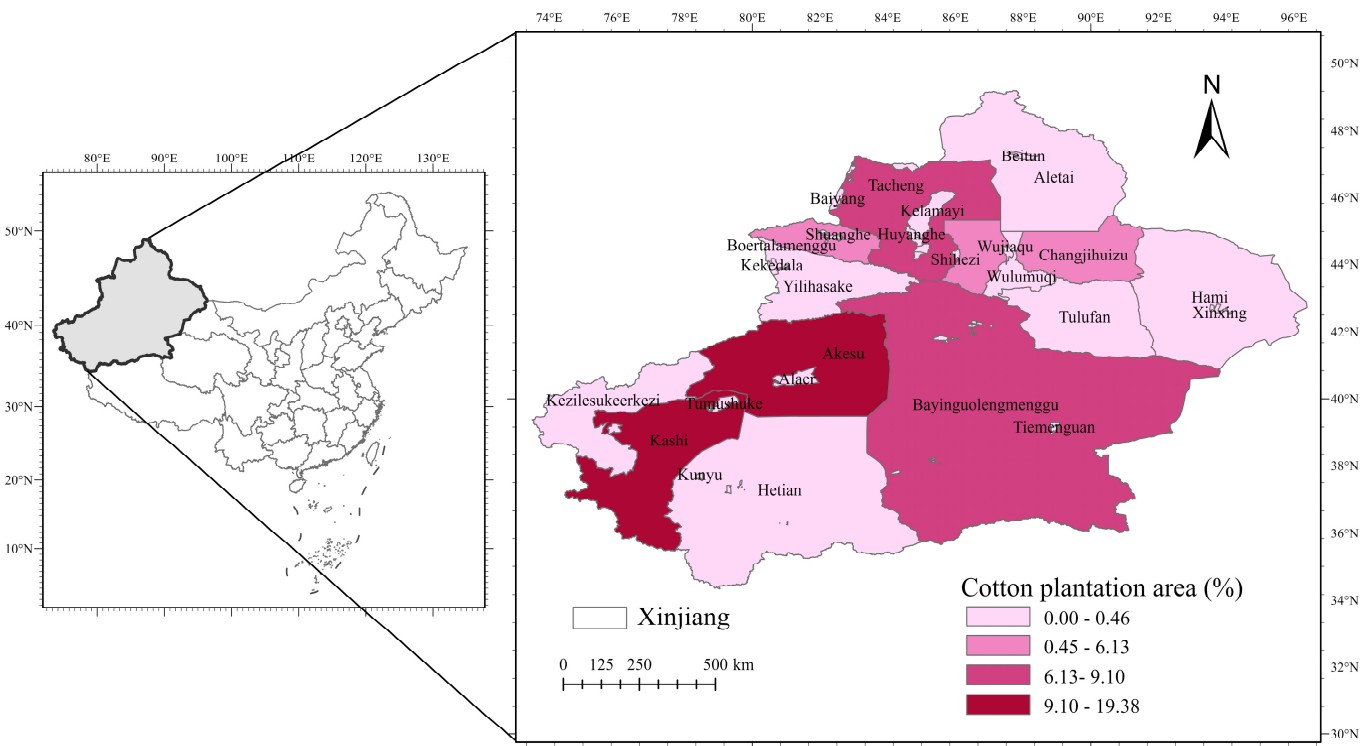

**Figure 4.** The proportion of cotton area planted in each city to the area of Xinjiang[9].

Fourthly, data from the China Rural Statistical Yearbook 2019 indicate that by the end of 2019, Xinjiang's usage of plastic mulch film constituted 17.6% of China's total usage, and its coverage area represented 20.13% of the national total, the highest in the country. Consequently, focusing on cotton farmers in Xinjiang, China, as the subject of this study, offers substantial research value in examining the impact of land tenure stability on the adoption of CCB.

The data for this study were primarily collected through a bilingual, face-to-face household survey of cotton farmers in Xinjiang, conducted by postgraduate students from Kashi University in 2019. The survey primarily focused on farmers' basic characteristics, family dynamics, land use, CCB adoption, and fertilizer usage. To enhance data accuracy and validity, the research team engaged experts for researcher training and carried out a preliminary survey in the Kashi and Changji areas of Xinjiang in August 2019 (refer to Figure 5). The sampling strategy combined stratified and random methods to ensure representativeness. Initially, the southern and northern regions of Xinjiang, including the corps areas, were selected to account for regional spatial variances. Subsequently, from these areas, two regions and one production and construction corps were chosen randomly based on descending order of cotton output. In each region, two counties were randomly selected following the principle of equidistant sampling of cotton yield. Within these counties, two towns each were randomly chosen, totaling 12 sample towns for the survey. Ultimately, out of the collected samples, 349 were deemed valid after excluding those with missing variables, resulting in a valid response rate of 94.17%.

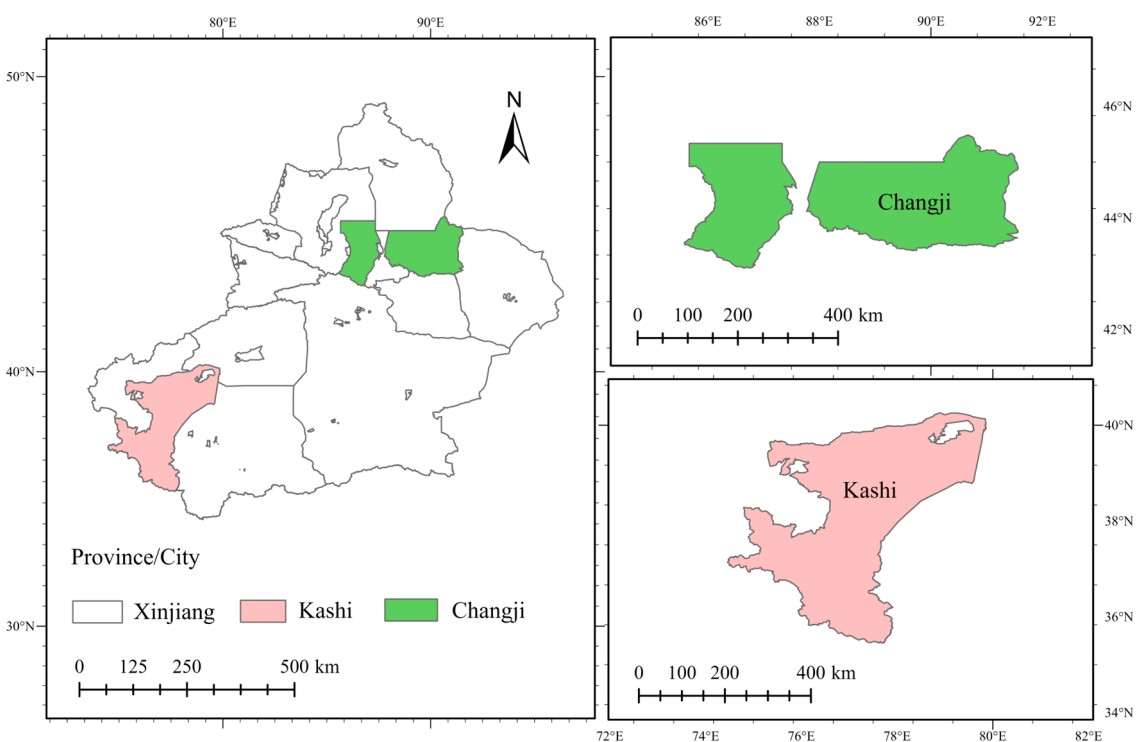

**Figure 5.** Field experiment sites.

*3.3. Specification of Independent Variables*

(a) Dependent variable: farmers' CCB adoption. Recognized as green agricultural technology, this study adopts the measurement approaches used by Li et al. [32] and Mao et al. [53] for assessing farmers' CCB adoption. The assessment includes three dimensions: (1) adoption decision—determined by whether farmers adopt CCB, with a value of 1 indicating adoption and 0 indicating non-adoption; (2) adoption extent—quantified by the proportion of CCB, calculated as the ratio of the CCB area to the total planting area (%); (3) adoption duration—measured by the cumulative years of farmers' engagement in CCB.

(b) Explanatory variables: land tenure stability, which encompasses land tenure duration and lease type. Land tenure duration is indicated by the number of years farmers have under land transfer contracts, with a longer duration signifying greater stability. According to Ma et al. [46], farmers with longer land tenure periods are more inclined towards green production practices compared to those with shorter tenures. Lease type is determined by the mode of contract signing. A written agreement, as opposed to verbal or no agreements, tends to foster long-term transactional relationships and enhances farmers' investment expectations in land.

(c) Control variables. This study includes various control variables that potentially influence farmers' green production behavior, as identified in previous research [65]. These include household and family characteristics, management practices, and soil quality indicators such as the age and gender of the household head, their educational level, farming experience, number of household farm laborers, technical subsidies, and historical cotton fertility. Research indicates that households led by younger, more educated males with extensive farming experience tend to positively impact CCB adoption [46,57]. Consequently, variables such as the age, gender, educational attainment, and cotton farming experience of the household head are integrated into the model. Additionally, findings suggest that a higher number of family laborers, increased technical subsidies, and poorer soil fertility correlate with a greater propensity towards CCB usage [4,32,66]. Thus, the model also incorporates the total number of family members, technical subsidy levels, and cotton soil fertility over the years. Furthermore, given the unique management system of Xinjiang corps, which differs markedly from local management systems in Xinjiang, the "corps

farmers'' variable is also controlled for in the model. Detailed definitions and descriptive statistics of these variables are presented in Table 1.

**Table 1.** Summary Statistics.

| Variable | Explanation | Mean | Std.dev | Min | Max |
|---|---|---|---|---|---|
| Explained variable | | | | | |
| CCB adoption | Whether farmers adopt CCB: Yes = 1; No = 0 | 0.762 | 0.426 | 0 | 1 |
| CCB adoption rate | Farmers' adoption proportion of CCB (%) | 61.488 | 40.904 | 0 | 100 |
| Adoption time of CCB | Farmers' cumulative adoption years of CCB (years) | 4.284 | 5.873 | 0 | 38 |
| Key explanatory variable | | | | | |
| Land tenure | Period for farmers to transfer cultivated land (years) | 6.407 | 4.575 | 0 | 30 |
| Lease type | No agreement = 0; oral = 1; written, but not completed by local government = 2; written and completed by the local government = 3 | 2.083 | 1.038 | 0 | 3 |
| Risk aversion | The degree of farmers' risk aversion($\sigma$) | 0.774 | 0.409 | 0.05 | 1.45 |
| Time preferences | Whether household head is far-sighted: Yes = 1; No = 0 | 0.095 | 0.293 | 0 | 1 |
| Technical training | Whether farmers participate in CCB training: Yes = 1; No = 0 | 0.397 | 0.490 | 0 | 1 |
| Farm scale | Actual cotton-planting area of farmers (hundred mu) | 4.477 | 14.480 | 0.032 | 160 |
| Control Variable | | | | | |
| Age | Age of household head (years) | 50.043 | 9.631 | 24 | 90 |
| Gender | Gender of household head: Male = 1; Female = 0 | 0.894 | 0.308 | 0 | 1 |
| Education | Education level of household head (years) | 7.977 | 2.794 | 0 | 16 |
| Cotton-planting experience | Cotton-planting years of household head (years) | 15.33 | 9.617 | 0 | 50 |
| Household total numbers | Total number of farmers' households (person) | 4.527 | 1.629 | 1 | 16 |
| Technical subsidy | Whether there is plastic mulch film subsidy locally: Yes = 1; No = 0 | 0.338 | 0.474 | 0 | 1 |
| Corps farmers | Whether farmers are corps farmers: Yes = 1; No = 0 | 0.249 | 0.433 | 0 | 1 |
| Cotton fertility over the years | Soil fertility of farmers' cotton fields: 1 = Poor; 2 = Medium; 3 = Good; 4 = Excellent | 2.309 | 0.759 | 1 | 4 |

Table 1 reveals that approximately 76.2% of the sampled farmers adopted CCB, with an average adoption rate of 61.488%. Among these, male-headed households comprised 89.4%, with the average age of household heads being 50.043 years and their average educational attainment spanning 7.977 years. The average duration of their planting experience was 15.33 years. From a land tenure perspective, the average tenure duration for the sampled farmers was 6.407 years, and the average cotton-planting area was 4.477 mu, predominantly featuring medium fertility. The average household size was 4.4527 members. Regarding production and operation characteristics, approximately 33.8% of the farmers received technical subsidies. Moreover, the sample indicated an average risk preference score of 0.774, with about 9.5% of farmers exhibiting a high degree of time preference.

## 4. Results

### 4.1. Benchmark Results

The Impact of Land Tenure Stability on Farmers' CCB Adoption

This paper examines the effect of land tenure stability on the adoption of CCB by farmers. Table 2 presents the benchmark results, with Columns 1 and 2 detailing the model's estimation of the impact of land tenure stability on the likelihood of farmers adopting CCB. The findings indicate a significantly positive correlation at the 1% level, where a 1% increase in land tenure correlates with a 2.1% rise in CCB adoption, thereby supporting hypothesis H1. This suggests that extended land tenure increases the probability of farmers adopting CCB. Columns 3 and 4 of Table 2 focus on the effect of land tenure stability on the adoption ratio of CCB among farmers, revealing a similarly positive association at the 1% significance level, further corroborating hypothesis H1. The underlying reasons might include the following: Firstly, stable land tenure serves as collateral for loans, facilitating

farmers' access to credit and technical funds, thus aiding CCB adoption. Secondly, longer land tenure enhances stability, encouraging long-term land investment under property rights incentives, leading to CCB adoption. Finally, stable land tenure increases farmers' motivation to participate in technical training, thereby improving their understanding of CCB principles and operations, enhancing green production awareness, and fostering receptivity to new technology.

**Table 2.** Estimated results of the influence of land tenure stability on farmers' CCB adoption.

| | (1) | (2) | (3) | (4) |
|---|---|---|---|---|
| | CCB Adoption | CCB Adoption | CCB Adoption Rate | CCB Adoption Rate |
| Land tenure | 0.019 *** | 0.021 *** | 1.835 *** | 1.961 *** |
| | (0.005) | (0.005) | (0.508) | (0.530) |
| Age | | −0.003 | | −0.325 |
| | | (0.003) | | (0.251) |
| Gender | | 0.028 | | 3.222 |
| | | (0.076) | | (7.444) |
| Education | | −0.007 | | −1.544 * |
| | | (0.008) | | (0.802) |
| Cotton-planting experience | | −0.000 | | −0.174 |
| | | (0.003) | | (0.241) |
| Household total numbers | | 0.008 | | −0.221 |
| | | (0.016) | | (1.408) |
| Technical subsidy | | 0.150 *** | | 5.351 |
| | | (0.042) | | (4.299) |
| Corps farmers | | −0.089 | | −1.054 |
| | | (0.056) | | (5.699) |
| Cotton fertility over the years | | −0.060 * | | −4.875 |
| | | (0.031) | | (2.992) |
| Constant | 0.639 *** | 0.900 *** | 49.871 *** | 88.178 *** |
| | (0.044) | (0.203) | (4.031) | (19.480) |
| Observation | 349 | 349 | 338 | 338 |
| $R^2$ | 0.044 | 0.107 | 0.043 | 0.068 |

Note: ***, * are significant at the level of 1% and 10%, respectively; standard errors are in brackets.

### 4.2. Robustness Check

To verify the robustness of the benchmark results, this study conducts tests along two dimensions: (1) Altering the core explanatory variables. Columns 1 and 2 in Table A1 (see Appendix A) substitute the core explanatory variable with "plastic mulch film recovery time", defined as "the cumulative number of years farmers have engaged in CCB". (2) Modifying the dependent variable. Columns 3 and 4 in Table A1 (see Appendix A) change the dependent variable to lease type. Across these variations, the results consistently indicate that the positive impact of land tenure stability on farmers' CCB adoption remains fundamentally unchanged despite the alterations in measurement methods of the dependent variables.

### 4.3. Endogeneity Problem

The analysis indicates that land tenure stability significantly enhances farmers' adoption of CCB. However, this conclusion may be subject to bias due to endogenous issues. Firstly, the problem of missing variables is notable. The current model may overlook certain unobserved factors influencing CCB adoption, leading to endogeneity. For instance, the model does not account for governmental economic interventions in agriculture, which can impact the market prices of agricultural products and subsequently affect CCB adoption [67]. Additionally, the model fails to consider the effect of CCB's technical complexity on its adoption, where higher complexity might hinder uptake [47]. Hence, the "missing variables" issue is present. Secondly, there is the concern of reverse causality. While CCB can yield economic benefits and enhance farmers' income, necessitating long-term and

stable land tenure for property rights protection, this could influence the demand for land tenure stability among farmers. Therefore, the possibility of "reverse causality" cannot be entirely discounted in this study.

This study employs instrumental variables (IVs) to address the issue of endogeneity. Following the criteria established by Imbens [68], the chosen IV should be correlated with the endogenous variable but not with the error term. Therefore, drawing upon the IV selection approach of Fisman and Svensson [69] and Ren et al. [62], this study selects the average land tenure of other villagers in the same village as the IV. Theoretically, this average land tenure influences farmers' land tenure decisions while remaining unrelated to their CCB adoption, thereby fulfilling the requirements for correlation and exogeneity. The IV results, presented in Table A2 in Appendix A, show that the land tenure coefficient is significantly positive at both the 5% and 1% levels. This indicates a substantial role of land tenure in enhancing farmers' CCB adoption, aligning with the benchmark findings. Furthermore, a weak instrumental variable test was conducted. The first-stage F statistic of 54.53 significantly exceeds the Cragg–Donald critical value [70], affirming the appropriateness of using the average land tenure of other villagers as the IV for land tenure in the same village, and negating any concerns regarding weak instrumental variables.

*4.4. Mechanism Analysis*

Land tenure stability facilitates farmers' adoption of CCB via land mortgage loans. Stable land tenure, evidenced by long-term land contract rights certification, empowers farmers to diminish credit evaluation costs with banks and thereby secure credit funds through land mortgages [71]. These credit funds, in turn, increase the likelihood of farmers making protective investments in their land [44], which further encourages CCB adoption. Consequently, it can be inferred that land tenure stability impacts farmers' access to land mortgage loans, and these loans significantly boost farmers' CCB adoption. This section aims to examine whether land tenure stability influences farmers' CCB adoption through the mechanism of land mortgage loans. The model employed is delineated as follows:

$$Landmortgage = \alpha + \beta Tenure + \gamma X + \mu_1 \tag{2}$$

$$Y_1 = \alpha + \beta Landmortgage + \gamma X + \mu_1 \tag{3}$$

$$Y_1 = \alpha + \beta Tenure + \varepsilon Landmortgage + \gamma X + \mu_1 \tag{4}$$

In this analysis, *Landmortgage* represents the behavior of land mortgage loans, determined by whether farmers engage in such loans. The definition of other variables aligns with that in Formula (1). Table 3 presents the results: Column (1) corroborates the benchmark regression findings. Column (2) examines the effect of land mortgage loans on CCB adoption, indicating that a higher likelihood of land mortgage loans substantially enhances the adoption of CCB. Column (3) explores the impact of land tenure stability on land mortgage loans, revealing a significant positive influence. In Column (4), both land tenure stability and land mortgage loans are considered. Here, the coefficient for land mortgage loans is notably positive at the 10% significance level, while the coefficient for land tenure stability remains positive. The mediation effect analysis indicates percentages of 4.59% and 5.12%, suggesting that land mortgage loans serve as a partial mediator. Thus, land tenure stability influences farmers' CCB adoption through land mortgage loans. These findings align with the paper's hypothesis that land tenure stability enhances farmers' propensity to adopt CCB via land mortgage loans.

**Table 3.** The mechanism test results of the influence of land tenure stability on farmers' CCB adoption.

**Panel A CCB Adoption**

|  | (1) | (2) | (3) | (4) |
|---|---|---|---|---|
|  | CCB Adoption | CCB Adoption | Land Mortgage Loans | CCB Adoption |
| Land tenure | 0.021 *** |  | 0.009 ** | 0.020 *** |
|  | (0.005) |  | (0.005) | (0.006) |
| Land mortgage loans |  | 0.142 ** |  | 0.106 * |
|  |  | (0.061) |  | (0.060) |
| Control variables | Yes | Yes | Yes | Yes |
| Constant | 0.900 *** | 0.943 *** | 0.254 | 0.874 *** |
|  | (0.203) | (0.205) | (0.163) | (0.201) |
| Observation | 349 | 349 | 349 | 349 |
| $R^2$ | 0.107 | 0.069 | 0.080 | 0.113 |

**Panel B CCB Adoption Rate**

|  | (1) | (2) | (3) | (4) |
|---|---|---|---|---|
|  | CCB Adoption Rate | CCB Adoption Rate | Land Mortgage Loans | CCB Adoption Rate |
| Land tenure | 1.961 *** |  | 0.009 ** | 1.860 *** |
|  | (0.530) |  | (0.005) | (0.536) |
| Land mortgage loans |  | 14.654 ** |  | 11.368 * |
|  |  | (6.464) |  | (6.338) |
| Control variables | Yes | Yes | Yes | Yes |
| Constant | 88.178 *** | 91.679 *** | 0.254 | 85.018 *** |
|  | (19.480) | (19.852) | (0.163) | (19.363) |
| Observation | 349 | 349 | 349 | 349 |
| $R^2$ | 0.068 | 0.035 | 0.080 | 0.075 |

Note: ***, **, and * are significant at the level of 1%, 5%, and 10%, respectively; standard errors are in brackets.

### 4.5. Additional Analysis

4.5.1. Land Tenure Stability and Risk Aversion on Farmers' CCB Adoption

This study investigates the moderating role of land tenure stability on the impact of risk aversion on farmers' adoption of CCB. Specifically, it aims to determine whether land tenure stability can positively mitigate the inhibitory effect of risk aversion on CCB adoption. The hypothesis posited is that a more substantial moderating effect correlates with increased farmer enthusiasm for adopting CCB.

Table 4 displays the findings. Columns 1 and 2 detail the moderating influence of land tenure stability on risk aversion regarding the adoption of CCB. Columns 3 and 4 explore this effect on the adoption ratio of CCB among farmers. Across all results, it is evident that land tenure stability positively moderates the deterrent effect of risk aversion on the adoption ratio of CCB. This outcome may be attributed to frequent agricultural risks, such as degradation in cultivated land quality, which substantially hinder farmers' income and the realization of green agricultural development goals. Typically, more risk-averse farmers are less inclined to adopt CCB. Enhanced land tenure stability, signifying assured land investment benefits, motivates farmers to undertake adaptive measures against risks. CCB contributes to mitigating agricultural risks and improving soil quality. Consequently, as land tenure duration increases, risk-averse farmers are more likely to utilize CCB, thus corroborating hypothesis H2.

**Table 4.** Estimated results of the influence of land tenure stability and risk aversion on farmers' CCB adoption.

| | (1) | (2) | (3) | (4) |
|---|---|---|---|---|
| | **CCB Adoption** | **CCB Adoption** | **CCB Adoption Rate** | **CCB Adoption Rate** |
| Land tenure | 0.019 *** | 0.012 * | 1.877 *** | 0.968 |
| | (0.005) | (0.006) | (0.544) | (0.622) |
| Risk aversion | −0.116 ** | −0.270 *** | −3.628 | −24.839 *** |
| | (0.056) | (0.098) | (5.662) | (9.125) |
| Land tenure × risk aversion | | 0.025 ** | | 3.510 *** |
| | | (0.011) | | (1.093) |
| Control variables | Yes | Yes | Yes | Yes |
| Constant | 0.951 *** | 0.996 *** | 89.632 *** | 96.042 *** |
| | (0.208) | (0.209) | (19.752) | (20.060) |
| Observation | 349 | 349 | 338 | 338 |
| $R^2$ | 0.118 | 0.129 | 0.069 | 0.093 |

Note: ***, **, and * are significant at the level of 1%, 5%, and 10%, respectively; standard errors are in brackets.

### 4.5.2. Land Tenure Stability and Time Preferences on Farmers' CCB Adoption

This study further examines the moderating role of land tenure stability on the impact of time preferences on farmers' adoption of CCB. Using the duration of land tenure as a moderator, the study assesses whether the influence of time preferences on CCB adoption is contingent upon land tenure stability. It posits that a more pronounced moderating effect correlates with increased farmer enthusiasm for CCB adoption.

Table 5 displays the findings. The interaction between time preferences and land tenure stability, as shown in Column 2, is significantly positive at the 1% level. This suggests that land tenure stability effectively mitigates the negative impact of time preferences on the likelihood of farmers adopting CCB. Hence, greater land tenure stability is associated with a reduced inhibitory effect of time preferences on CCB adoption. This relationship can be attributed to the influence of time preferences on decision making regarding CCB adoption, where a higher degree of time preference is linked to a lower adoption rate. Enhanced land tenure stability, indicative of long-term investment encouragement, likely weakens the adverse effect of time preferences on CCB adoption. These insights align with hypothesis H3.

**Table 5.** Estimated results of the influence of land tenure stability and time preferences on farmers' CCB adoption.

| | (1) | (2) | (3) | (4) |
|---|---|---|---|---|
| | **CCB Adoption** | **CCB Adoption** | **CCB Adoption Rate** | **CCB Adoption Rate** |
| Land tenure | 0.017 *** | 0.013 ** | 1.622 *** | 1.420 *** |
| | (0.005) | (0.005) | (0.525) | (0.547) |
| Time preferences | −0.406 *** | −0.644 *** | −31.534 *** | −42.459 *** |
| | (0.082) | (0.106) | (7.921) | (10.328) |
| Land tenure × time preferences | | 0.062 *** | | 2.858 |
| | | (0.012) | | (1.806) |
| Control variables | Yes | Yes | Yes | Yes |
| Constant | 0.907 *** | 0.913 *** | 88.622 *** | 88.899 *** |
| | (0.182) | (0.175) | (17.866) | (17.762) |
| Observation | 349 | 349 | 338 | 338 |
| $R^2$ | 0.178 | 0.205 | 0.116 | 0.122 |

Note: *** and ** are significant at the level of 1% and 5%, respectively; standard errors are in brackets.

### 4.5.3. Land Tenure Stability and Farmers' CCB Adoption: Farm Scale Differences

The variation in farmers' adoption of CCB according to farm size is a key focus of this study. Specifically, it examines how land tenure stability affects CCB adoption across

different farm scales. Table 6 presents the results of this analysis. Columns 1 and 2 detail the relationship between land tenure stability and CCB adoption among varying farm sizes. Columns 3 and 4 explore how land tenure stability influences the CCB adoption ratio across different farm scales. The findings consistently indicate that, in comparison to small-scale farmers, land tenure stability exerts a more substantial positive impact on CCB adoption among large-scale farmers. This can be attributed to several factors: Large-scale farmers generally prioritize long-term income and are more inclined to implement technologies with prolonged income benefits, such as CCB. Furthermore, by expanding their farm size, they can potentially reduce production costs and achieve economies of scale, thereby enhancing their willingness to adopt CCB.

**Table 6.** The influence of land tenure stability on farmers' CCB adoption: farm scale difference.

| | (1) | (2) | (3) | (4) |
|---|---|---|---|---|
| | **CCB Adoption** | | **CCB Adoption Rate** | |
| | **Large-Scale Farmers** | **Small-Scale Farmers** | **Large-Scale Farmers** | **Small-Scale Farmers** |
| Land tenure | 0.035 *** | 0.018 ** | 2.886 *** | 1.724 ** |
| | (0.009) | (0.008) | (0.859) | (0.741) |
| Control variables | Yes | Yes | Yes | Yes |
| Constant | 0.432 | 1.171 *** | 51.299 | 113.358 *** |
| | (0.347) | (0.246) | (31.312) | (24.510) |
| Observation | 154 | 195 | 150 | 188 |
| $R^2$ | 0.244 | 0.120 | 0.179 | 0.091 |

Note: *** and ** are significant at the level of 1% and 5%, respectively; standard errors are in brackets.

### 4.5.4. Land Tenure Stability and Farmers' CCB Adoption: Technical Training

This study explores how land tenure stability influences farmers' CCB adoption in relation to their participation in technical training. Table 7 presents the findings. Columns 1 and 2 detail the impact of land tenure stability on CCB adoption among farmers with varying levels of technical training. Columns 3 and 4 examine how land tenure stability affects the adoption ratio of CCB among farmers based on their technical training participation. The results consistently indicate that land tenure stability exerts a more substantial positive influence on CCB adoption among farmers who have participated in technical training compared to those who have not. This effect can likely be attributed to technical training serving as an information exchange platform. Such platforms facilitate the increased exchange of technical knowledge among farmers, enhancing their understanding of CCB and minimizing potential losses due to incomplete technical knowledge. Consequently, this increased mastery encourages a greater propensity to adopt CCB.

**Table 7.** The influence of land tenure stability on farmers' CCB adoption: technical training difference.

| | (1) | (2) | (3) | (4) |
|---|---|---|---|---|
| | **CCB Adoption** | | **CCB Adoption Rate** | |
| | **Participate in Technical Training** | **Not Participating in Technical Training** | **Participate in Technical Training** | **Not Participating in Technical Training** |
| Land tenure | 0.035 *** | 0.015 ** | 2.994 *** | 1.544 ** |
| | (0.008) | (0.007) | (1.072) | (0.665) |
| Control variables | Yes | Yes | Yes | Yes |
| Observation | 122 | 185 | 122 | 185 |
| $R^2$ | 0.325 | 0.109 | 0.207 | 0.067 |

Note: *** and ** are significant at the level of 1% and 5%, respectively; standard errors are in brackets.

## 5. Conclusions

Amid concerns about cultivated land quality degradation and severe agricultural non-point-source pollution, this study, utilizing 2019 survey data from cotton farmers in Xinjiang, examines the impact of land tenure stability on CCB adoption. The key findings include the following: Firstly, land tenure stability significantly enhances the likelihood of CCB adoption among farmers, a trend confirmed by robustness checks. Secondly, land tenure stability mitigates the adverse effects of risk aversion and time preferences on CCB adoption, thus addressing the low adoption rates linked to these factors. Thirdly, the effect of land tenure stability on CCB adoption varies according to farm scale and technical training, with a more pronounced positive impact observed among larger-scale farmers and those participating in technical training.

This study examines the influence of land tenure stability on CCB from the perspective of agricultural films. While many studies have investigated the impact of land tenure stability on the adoption of various farming techniques such as the use of organic fertilizers and agrochemicals, the diverse nature of farming techniques suggests that the effects of land tenure stability may vary across different agricultural film technologies [3,72,73]. Therefore, we focus on the agricultural film perspective to validate the impact of land tenure stability on CCB, thus complementing the existing literature and reinforcing the conclusion that land tenure stability contributes to enhancing CCB.

This research contributes to the existing body of knowledge on factors influencing CCB adoption among farmers, offering new insights into the role of land tenure stability, previously underexplored in relation to its effects via land mortgage loans and its moderating influence on risk aversion and time preferences. Furthermore, the study's findings have significant implications for green agricultural development and "white pollution" governance in developing countries.

Policy recommendations based on these findings are as follows: Firstly, governments should address land tenure instability, establish transparent land rights, and support village autonomy and oversight to ensure long-term land security for farmers. Secondly, policy measures should include technical subsidies and credit access, coupled with efforts to strengthen land tenure stability. Thirdly, initiatives to encourage moderate farm scaling and enhanced land transfer systems are vital, with particular emphasis on supporting large-scale farmers. Lastly, the importance of technical training and localized CCB promotion to elevate farmers' agricultural practices is underscored.

This study, however, is not without limitations. Primarily, it relies on survey data from 349 cotton farmers in Xinjiang, and although the findings are robust, broader data from various provinces could enhance their universality. Future research should also utilize panel data to better capture the dynamic effects of land tenure stability on CCB adoption. Additionally, subsequent studies might expand the scope to include other green agricultural technologies beyond CCB.

**Author Contributions:** Data curation, J.L.; Writing—original draft, G.L.; Writing—review & editing, X.Z. All authors have read and agreed to the published version of the manuscript.

**Funding:** This research was funded by the Humanities and Social Science Research General Project of the Ministry of Education of China (21XJC790008), and the Shaanxi Social Science Foundation (2021D028).

**Institutional Review Board Statement:** This study was granted exemption by the Ethical Review Committee of the Institute of Central Asian Studies, Shaanxi Normal University (approval number: 2018-011-02). We certify that the study was performed in accordance with the 1964 declaration of HELSINKI and later amendments. Ethical review and approval were waived for this study due to REASON: This survey was conducted with the assurance of anonymity, and all participants were fully informed of the reasons for conducting this survey study, the use of relevant data, etc. Although this survey is primary data, no personal information of the participants (e.g., name, phone number, address, etc.) was recorded during the survey process, which makes it impossible to trace back to

the participants themselves, and it is unlikely that this survey will cause any psychological harm to the participants.

**Data Availability Statement:** The original contributions presented in the study are included in the article, further inquiries can be directed to the corresponding author.

**Conflicts of Interest:** The authors declare no conflict of interest.

## Appendix A

**Table A1.** Robustness results of the influence of land tenure stability on farmers' adoption behavior of CCB.

| | (1) | (2) | (3) | (4) |
|---|---|---|---|---|
| | **Adoption Time of CCB** | **Adoption Time of CCB** | **CCB Adoption** | **CCB Adoption Rate** |
| Land tenure | 0.166 ** | 0.201 *** | | |
| | (0.065) | (0.073) | | |
| Lease type | | | 0.062 *** | 7.083 *** |
| | | | (0.024) | (2.300) |
| Age | | −0.078 ** | −0.003 | −0.338 |
| | | (0.031) | (0.003) | (0.258) |
| Gender | | 0.775 | -0.011 | -0.928 |
| | | (0.851) | (0.074) | (7.294) |
| Education | | −0.226 * | −0.003 | −1.058 |
| | | (0.130) | (0.008) | (0.816) |
| Cotton planting experience | | 0.141 *** | −0.001 | −0.208 |
| | | (0.045) | (0.003) | (0.247) |
| Household total numbers | | −0.191 | 0.006 | −0.409 |
| | | (0.153) | (0.017) | (1.478) |
| Technical subsidy | | −1.019 | 0.190 *** | 9.593 ** |
| | | (0.683) | (0.042) | (4.381) |
| Corps farmers | | 1.663 ** | −0.059 | 1.928 |
| | | (0.840) | (0.057) | (5.743) |
| Cotton fertility over the years | | -0.087 | -0.034 | -2.195 |
| | | (0.457) | (0.032) | (3.094) |
| Constant | 3.232 *** | 6.821 ** | 0.852 *** | 79.274 *** |
| | (0.455) | (2.814) | (0.213) | (20.938) |
| Observation | 335 | 335 | 349 | 338 |
| $R^2$ | 0.017 | 0.085 | 0.080 | 0.053 |

Note: ***, **, and * are significant at the level of 1%, 5%, and 10%, respectively; standard errors are in brackets.

**Table A2.** The IV estimated results of the influence of land tenure stability on farmers' adoption behavior of CCB.

| | (1) | (2) | (3) |
|---|---|---|---|
| | **First Stage** | **CCB Adoption** | **CCB Adoption Rate** |
| Land tenure | | 0.027 ** | 3.533 *** |
| | | (0.012) | (1.290) |
| IV | 0.999 *** | | |
| | (0.135) | | |
| Constant | | 0.877 *** | 81.676 *** |
| | | (0.206) | (19.939) |
| Observation | 349 | 349 | 338 |
| $R^2$ | | 0.103 | 0.039 |

Note: *** and ** are significant at the level of 1% and 5%, respectively; standard errors are in brackets.

## Notes

1.  "White pollution" refers to the environmental degradation caused by the accumulation of waste plastic products. This term particularly highlights the issue of plastic pollutants, such as agricultural mulch films made from polymers including polystyrene, polypropylene, and polyvinyl chloride. The environmental challenge posed by these materials stems from their resistance to degradation. After their use, these plastics are frequently discarded, contributing to the growing problem of solid waste pollution.

2.  Source: China Rural Statistical Yearbook in 2023.

3.  Property right plays a role in stabilizing the main body practical expectation through the main body possess, use, benefit and punish rights (Barzel, 1997).

4.  The data comes from the Ministry of Agriculture and Rural Affairs of the People's Republic of China, "the Bulletin of 2019 National Cultivated Land Quality Grades". http://www.moa.gov.cn/xw/zwdt/202005/t20200512_6343750.htm, accessed on 2 February 2024.

5.  The data comes from the Ministry of Agriculture and Rural Affairs of the People's Republic of China, "the Bulletin of 2019 National Cultivated Land Quality Grades".

6.  The data comes from Agriculture and Rural Department of Xinjiang Uygur Autonomous Region, "Further Improvement of Moderate Scale Management Level of Xinjiang Agriculture".

7.  The data comes from the Announcement of National Bureau of Statistics on Cotton Production in 2019. http://www.stats.gov.cn/tjsj/zxfb/201912/t20191217_1718007.html, accessed on 2 February 2024.

8.  The source of the data for making the map is "China Rural Statistical Yearbook" in 2020.

9.  The source of the data for making the map is "Xinjiang Provincial Statistical Yearbook" in 2020.

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
