# Peer review of "Land Tenure, Loans, and Farmers’ Cropland Conservation Behavior: Evidence from Rural Northwest China"

_land, doi:10.3390/land13040413_

Round 1

Reviewer 1 Report

Comments and Suggestions for Authors

The protection and utilization of cultivated land is related to human life and health, which has been the focus of attention from all walks of life. Based on the survey data of cotton farmers in Xinjiang, from the perspective of land ownership stability, the author pays attention to the influence of land ownership on mulch recycling, and examines the regulatory mechanism. In general, the topic selection has certain significance, the research design is relatively reasonable, and the conclusions are basically in line with common sense. In order to better improve the quality of the paper, several suggestions for reference:

       (1) The review in the introduction is relatively simple. At present, there is a lack of systematic review of research on mulch use, especially the comparison of land ownership, loan, risk and time preference with research on mulch use or farmland protection. As far as I know, the topic studied by the author is not a new topic in China, and there should be some similar research. The author should put his paper into the literature to have a dialogue, so as to enhance the persuasion of the marginal contribution of the research. We can't just mention that there is little research looking at land tenure stability.

       (2) The maps in FIG. 2 and FIG. 3 are somewhat deformed and do not conform to the map usage specifications of the Ministry of Natural Resources of China. It is suggested that the author download the base map from the official website of the Ministry of Natural Resources of China and redraw the two maps. If it is absolutely impossible, not having these two pictures will not affect the reading of the whole paper.

       (3) The section of theoretical analysis and research hypothesis suggests to draw a frame diagram of theoretical analysis to help readers better understand the logic of the author's research hypothesis.

       (4) In the part of research results, it is suggested to put out the table of results related to endogenous treatment.

       (5) This study lacks in-depth discussion and analysis, so it is suggested to add a discussion section to systematically discuss the differences between this study and similar studies and the reasons for the differences. At the same time, it can also point out the shortcomings of this study and the direction of further improvement in the future.

Comments on the Quality of English Language

Minor editing of English language required.

Reviewer 2 Report

Comments and Suggestions for Authors

This research study investigates the influence of land tenure stability on the adoption of Conservation Cotton Farming (CCB) among cotton farmers in Xinjiang, using survey data from 2019. The backdrop for this inquiry is concerns about the deterioration of cultivated land quality and the prevalence of non-point source pollution in agriculture. The key findings highlight that land tenure stability significantly increases the likelihood of CCB adoption among farmers. The stability of land tenure is observed to mitigate the negative effects of risk aversion and time preferences on CCB adoption, addressing the low adoption rates associated with these factors.

Furthermore, the research explores how the impact of land tenure stability on CCB adoption varies based on farm scale and technical training. Larger scale farmers and those involved in technical training experience a more pronounced positive impact. The study contributes novel insights to the existing knowledge on factors influencing CCB adoption, particularly shedding light on the role of land tenure stability, an aspect that has been previously underexplored, especially in relation to land mortgage loans.

The research underscores the significance of its findings for green agricultural development and pollution control in developing countries. In terms of policy recommendations, the study suggests that governments should prioritize addressing land tenure instability, ensuring transparent land rights, and supporting village autonomy to secure long-term land tenure for farmers. Additional recommendations include implementing technical subsidies and credit access measures, strengthening land tenure stability, and encouraging moderate farm scaling and improved land transfer systems, with a focus on supporting larger-scale farmers. The study also emphasizes the importance of technical training and localized CCB promotion to enhance farmers' agricultural practices.

However, the study acknowledges its limitations, primarily relying on survey data from 349 cotton farmers in Xinjiang. To enhance the universality of the findings, the researchers recommend broader data collection from various provinces and propose the use of panel data in future research to better capture the dynamic effects of land tenure stability on CCB adoption. Additionally, the researchers suggest expanding the scope of subsequent studies to include other green agricultural technologies beyond CCB.

I agree to the publication this article.

Round 2

Reviewer 1 Report

Comments and Suggestions for Authors

I have no other comments, thank you.